# Potential of Antimicrobial Peptide-Overexpressed *Tenebrio molitor* Larvae Extract as a Natural Preservative for Korean Traditional Sauces

**DOI:** 10.3390/insects13040381

**Published:** 2022-04-13

**Authors:** Dooseon Hwang, Seung Hun Lee, Tae-Won Goo, Eun-Young Yun

**Affiliations:** 1Department of Integrative Biological Sciences and Industry, Sejong University, Seoul 05006, Korea; h.michael8837@gmail.com (D.H.); g-d-lsh@hanmail.net (S.H.L.); 2Department of Biochemistry, College of Medicine, Dongguk University, Gyeongju 38766, Korea; gootw@dongguk.ac.kr

**Keywords:** antimicrobial peptide, aflatoxin, biofilm, food-poisoning bacteria, insect extract

## Abstract

**Simple Summary:**

In this study, the antimicrobial activity and stability of immunized *Tenebrio molitor* larvae extract (iTME) were confirmed, and the antibacterial activity was observed in Korean traditional soy sauce. As a result, iTME showed antibacterial activity against food poisoning bacteria and harmful fungi. It had stability in various pH, temperature, and salinity ranges, and antibacterial activity in Korean traditional soy sauce inoculated with food poisoning bacteria. Therefore, iTME has the potential as a natural preservative with stable antimicrobial activity.

**Abstract:**

Here, we aimed to produce a natural food preservative using a crude extract from edible, immunized *Tenebrio molitor* larvae (iTME), injected with edible bacteria using an edible solvent. Results showed that iTME had concentration-dependent inhibitory activity against food-poisoning bacteria *Escherichia coli*, *Bacillus cereus*, and *Staphylococcus aureus*, as well as against harmful fungi *Aspergillus flavus*, *Aspergillus parasiticus*, and *Pichia anomala*. Moreover, iTME showed antimicrobial activity against beneficial microorganisms *Bacillus subtilis* and *Aspergillus oryzae*, but not *Lactobacillus acidophilus*. Furthermore, the minimum inhibitory concentration of iTME against *E. coli*, *B. cereus*, and *S. aureus* was 1 mg/mL, and iTME did not lose its inhibitory activity when treated at varying temperature, pH, and salinity. In addition, the antibacterial activity was lost after reacting the iTME with trypsin and chymotrypsin. The addition of iTME to Ganjang inoculated with harmful bacteria inhibited bacterial growth. Therefore, we propose that iTME can be used as a safe natural preservative to prolong food shelf life by inhibiting the growth of food-poisoning bacteria in a variety of foods, including traditional sauces.

## 1. Introduction

Korean traditional sauces prepared from soybeans include soy sauce (Ganjang), soybean paste (Doenjang), and red pepper paste (Gochujang). Meju, which is the origin of these sauces, is traditionally prepared as follows: after boiling the soybean, it is shaped into a cuboid, and then it is hung in a well-ventilated place for four to five months. In this process, various microorganisms such as *Aspergillus* sp. and *Bacillus* sp. attach to and ferment Meju. Fermented Meju is immersed in brine for two to three months, and then the solid and liquid phases are separated that become Ganjang and Doengjang, respectively. Meanwhile, Gochujang is prepared by evenly mixing Meju, red pepper powder, and glutinous rice powder and ripening them [1].

*Aspergillus oryzae* breaks down the nutrients in Meju into an easily absorbable form [2]. However, contamination by aflatoxin-producing *Aspergillus flavus* and *Aspergillus parasiticus*, ochratoxin A-producing *Aspergillus ochraceus* and *Penicillium verrucosum*, and biogenic amine-producing *Escherichia* sp., *Salmonella* sp., *Pediococcus* sp., and *Streptococcus* sp. may occur under the natural fermentation process [3,4]. To solve this problem, koji made by artificially inoculating *Aspergillus oryzae* is used; however, it renders an inferior taste and nutrition compared to traditional Meju, which is naturally fermented by the complex action of various microorganisms, such as *Aspergillus* sp., *Bacillus* sp., *Rhizopus* sp., and *Mucor* sp. [5]. Even chemical preservatives used to inhibit harmful microorganisms do not have selective inhibitory activity against harmful microorganisms and are not easy to use in food due to consumer rejection [6]. Therefore, it is necessary to develop a preservative derived from a natural material that can suppress the growth of harmful microorganisms without reducing the original taste and aroma of Korean traditional sauces.

Antimicrobial peptides (AMPs) have an inhibitory effect on bacteria, fungi, viruses, and parasites, and thus play an important role in the innate immune system of eukaryotes and prokaryotes [7]. AMPs aggregate, penetrate the cell membrane, and form channels, resulting in cytoplasmic leakage. In severe cases, AMPs induce cell membrane disruption, leading to cell death [8,9]. Owing to these properties, AMPs do not generate resistant strains unlike conventional antibiotics, which develop resistance when specific target sites are changed or metabolic processes are altered [10]. Currently, AMPs are used in various fields such as antibiotics, anticancer drugs, cosmetic materials, and food [11,12,13,14]. AMPs used in food include nisin and pediocin isolated from *Lactococcus lactis* and *Pediococcus* sp., respectively, both of which are used for preserving dairy products [11,12,15]. However, using a single AMP derived from bacteria takes a lot of time and is expensive due to purification and synthesis. Moreover, the taste and flavor deteriorate when AMP-containing bacteria are directly added to food. In addition, a single AMP has a narrow spectrum of target bacteria [16,17].

Insects produce a variety of AMPs for defense against external pathogens, and they harbor more than 150 different AMP genes [18,19,20,21]. AMPs from insects have been applied in various fields, such as molecular breeding of pathogenic fungal-resistant transgenic crops [22,23,24,25], feed manufacturing for rainbow trout and broilers [26,27], and production of natural preservatives for food packaging [28]. To date, no studies have reported the direct application of insect-derived AMPs to food, since insects have been officially recognized as new raw food materials only recently, and interest in edible insects is increasing worldwide [29,30]. Therefore, in this study, we utilized insect-derived AMPs as a sauce preservative. An extract containing AMPs, which exhibits a wide range of antibacterial activity and is inexpensive, was prepared using *Tenebrio molitor* larvae (mealworm, TML), an edible insect. Since the preparation of Meju with TML has been previously reported [31], we assumed that this extract would not deteriorate the taste of Ganjang. Finally, to confirm the potential of immunized *Tenebrio molitor* larvae extract (iTME) as a sauce preservative, we analyzed the antimicrobial activity and stability of iTME.

## 2. Materials and Methods

### 2.1. Tenebrio molitor Larvae, Fungal, and Bacterial Strains

The last-instar TML, an edible insect recognized by the Ministry of Food and Drug Administration in Cheongju-si, Chungcheongbuk-do, Korea, was purchased from Michin Mealworm (Yongin-si, Gyeonggi-do, Korea). All strains used in this study were obtained from the American Type Culture Collection (ATCC), Manassas, VA, USA, the Korean Agricultural Collection (KACC), Wanju-gun, Jeollabuk-do, Korea, and the Korea Culture Center of Microorganisms (KCCM), Seoul, Korea. The bacterium used to induce AMP was *Lactiplantibacillus plantarum* (KACC 18510), an edible bacterium. The antimicrobial activity of iTME against harmful microorganisms was confirmed using *E. coli* (KCCM 11234), *S. aureus* (KCCM 11593), and *B. cereus* (ATCC 11778), which are all food poisoning bacteria registered in the Food Code [30], *Pichia anomala* (KACC 46786, forming the biofilm on the Korean traditional sauce), *A. flavus* (KACC 44983), and *A. parasiticus* (KACC 40074), an aflatoxin-producing fungus [32,33,34]. The inhibitory activity of iTME against beneficial microorganisms was confirmed using *B. subtilis* subsp. *subtilis* (KACC 15940), isolated from fermented Meju and Korean traditional sauces; *A. ozyzae* (KACC 44823), which is used to make Meju; and *Lactobacillus acidophilus* (KACC 12419), a human enterobacterium.

### 2.2. Preparation of TML Extract Containing Overexpressed AMPs Induced by Bacterial Challenge

Ten microliter (1 × 10^10^ colony-forming units (CFU)/mL) of *Lactiplantibacillus plantarum* (KACC 18510) was injected into the side of the abdomen of the TML using a sanitized insulin syringe to induce the production of AMPs. Subsequently, the AMP-induced TML (iTML) was placed at 25 °C for 6, 12, 24, 36, 48, or 72 h without feeding. After each period, the TML was rapidly frozen using liquid nitrogen and then ground in a mortar.

Meanwhile, iTME was prepared by mixing iTML powder with 20% glacial acetic acid (AcOH, Samyang industry, Ansan-si, Gyeonggi-do, Korea), an edible solvent, at a ratio of 1:10 (*w*/*v*), immersing it at 25 °C for 48 h, and then centrifuging at 8935× *g* for 30 min. The supernatant was filtered with a 0.45 µm syringe filter (Corning, NY, USA), concentrated using a speed vacuum (EYELA, Tokyo, Japan) for 48 h, and the weight of the remaining pellet was measured. The concentrated extract was resuspended in 0.05% AcOH and used for antimicrobial activity assays.

### 2.3. Quantitative Real-Time Polymerase Chain Reaction (qRT-PCR) for the Analysis of AMP Expression in iTML

Total RNA was isolated from iTML powder, which was prepared as described in Section 2.2., using TRIzol ^®^ Reagent (Invitrogen, Carlsbad, CA, USA). Subsequently, cDNA was synthesized with the Quantinova ^®^ reverse transcription kit (Qiagen, Hilden, Germany). The transcripts of AMPs (attacin, defensin, tenecin, and coleoptericin) in iTML were analyzed using the SensiFAST™ Sybr No-Rox Mix, 2× (Bioneer, Daejeon, Korea). Actin was used as an endogenous control, and relative gene expression levels were analyzed using the 2^−^^ΔΔCt^ method. Primers for *T. molitor* (Tm) ribosomal protein 27a (TmL27a) (forward: 5′-TCATCCTGAAGGCAAAGCTCCAGT-3′, reverse: 5′-AGGTTGGTTAGGCAGGCACCTTTA-3′), TmTenecin1 (forward: 5′-CAGCTGAAGAAATCGAACAAGG-3′, reverse: 5′-CAGACCCTCTTTCCGTTACAGT-3′), TmDefensin (forward: 5′-AAATCGAACAAGGCCAACAC-3′, reverse: 5′-GCAAATGCAGACCCTCTTTC-3′), TmColeoptericin (forward: 5′-GGACAGAATGGTGGATGGTC-3′, reverse: 5′-CTCCAACATTCCAGGTAGGC-3′), and TmAttacin (forward: 5′-GAAACGAAATGGAAGGTGGA-3′, reverse: 5′-TGCTTCGGCAGACAATACAG-3′) were used for cDNA amplification [35].

### 2.4. Radial Diffusion Assay (RDA) for the Antibacterial and Antifungal Activities of iTME

RDA against bacteria (*E. coli*, *S. aureus*, *B. cereus*, *B. subtilis*, and *Lactobacillus acidophilus*) and fungi (*A. flavus*, *A. parasiticus*, *P. anomala*, and *A. oryzae*) were conducted using iTME. First, 10 mL of underlay agar containing 100 mM sodium phosphate/citrate buffer, tryptic soy agar, distilled water was mixed with an inoculum of each bacterium or fungus. When the underlay agar solidified, a hole with a diameter of 3 mm was made, and 5 µL of the iTME (20 and 40 mg/mL) was added to the hole. After 3 h of incubation, 10 mL of overlay agar was added and incubated at 30 °C or 37 °C for 18 h.

### 2.5. Determination of the Minimum Inhibitory Concentration (MIC) of iTME against Food Poisoning Bacteria

Food-poisoning bacteria (*E. coli*, *S. aureus*, *B. cereus*) were inoculated in a liquid medium and cultured at 37 °C with shaking at 200 rpm to until 4 × 10^6^ CFU/mL. Bacterial cultures were then diluted to 1.0 × 10^6^ CFU/mL and added to wells in a 96-well plate. Bacteria were then treated with 0.8, 0.9, or 1 mg/mL iTME. After 18 h, the OD_600_ value was measured using a spectrophotometer (SPECTROstar^nano^, Ortenberg, Germany) to obtain the MIC of iTME against food-poisoning bacteria.

### 2.6. Stability of iTME

#### 2.6.1. Stability of iTME under High Temperature

To analyze the stability of the iTME at various temperatures, the MIC was measured at 40 °C, a high temperature that can be exposed during distribution, and at 100 °C for high temperature sterilization suggested in the Korean Food Code [30]. The iTME was incubated at 40 °C for 36, 48, and 72 h or heated at 100 °C for 5, 10, and 15 s. Thereafter, 1.0 × 10^6^ CFU/mL of *E. coli* was treated with 0.8, 0.9, or 1 mg/mL of iTME heated under the above conditions. After 18 h, the OD_600_ was measured.

#### 2.6.2. Stability of iTME under Various pH

To confirm the stability of iTME to pH, the pH of Ganjang and Gochujang were measured using a pH meter (LAQUA F-71, HORIBA, Kyoto, Japan). The pH of commercially available Ganjang and Gochujang (Mohyeon Traditional Fermented Soy Sauce, Universal Farm’s Meal, Sunchang-gun, Jeollabuk-do, Korea) was 4.74 ± 0.09 and 5.04 ± 0.07, respectively (Table 1). Then, the pH of these products were adjusted to pH 4, 5, or 6, and 1.0 × 10^6^ CFU/mL of *E. coli* was inoculated. Subsequently, 0.8, 0.9, or 1 mg/mL of iTME were mixed and cultured for 18 h. The MIC of iTME against *E. coli* was confirmed by measuring the OD_600_ value.

#### 2.6.3. Stability of iTME under Various Salinity

To apply iTME to Korean traditional sauces, the salinity of commercially available Ganjang and Gochujang (Mohyeon Traditional Fermented Soy Sauce, Universal Farm’s Meal) was measured using the Mohr method [1]. The salinities of commercially available Ganjang and Gochujang were 19.33 ± 1.68% and 6.96 ± 0.30%, respectively (Table 1). Then, the iTME was reacted in salinity values of 5, 7.5, 10, 15, 20, or 25% for 1 h, then 1.0 × 10^6^ CFU/mL of *E. coli* and 0.8, 0.9, or 1 mg/mL of iTME were inoculated and cultured. After 18 h, the OD_600_ was measured.

### 2.7. Analysis of the Antibacterial Activity of iTME under Human Proteolytic Enzymes

The antibacterial activity of iTME under two human intestinal proteolytic enzymes was analyzed. Trypsin (Sigma-Aldrich, St. Louis, MO, USA) and chymotrypsin (Sigma-Aldrich, St. Louis, MO, USA) derived from human pancreas were used as proteolytic enzymes. Five hundred microliters each of 50 nM trypsin and chymotrypsin and 100 µL of 1 mg/mL iTME were mixed. The mixture was incubated at 37 °C for 5, 10, 20, 30, and 60 min and then incubated in a water bath at 80 °C for 10 min to inactivate the proteolytic enzymes. Thereafter, 1.0 × 10^6^ CFU/mL of *E. coli*, *B. cereus*, and *S. aureus* were treated with 0.8, 0.9, or 1 mg/mL of iTME reacted with trypsin and chymotrypsin. After 18 h, the OD_600_ was measured.

### 2.8. Evaluation of the Preservative Potential of iTME for Ganjang

To evaluate the preservative potential of iTME in Ganjang, Ganjang was prepared without any preservative. Briefly, soybeans were soaked in water, boiled, and crushed, and then cubes were dried in a well-ventilated place for 7 d. Subsequently the cubes were tied with straw, hung for 40 d, dried, and fermented at 25–28 °C for 30 d. Then, the cubes were immersed in 20 L of water, in which 3 kg of sea salt was dissolved and left for 90 d. The cubes were then removed, and the liquid was used as Ganjang. Then, 1 × 10^5^ CFU/mL of *E. coli*, *S. aureus*, and *B. cereus* with 1 mg/mL of iTME were inoculated into 10 mL of Ganjang and incubated for 10 d at 4, 18, and 30 °C, respectively. After 1, 2, 4, 6, 8, and 10 d, samples were collected and inoculated into each selective medium (*E. coli*, McConkey agar; *S. aureus*, mannitol salt agar with egg yolk; *B. cereus*, mannitol egg yolk polymyxin agar). After incubation for 48 h, bacterial growth was observed.

### 2.9. Statistical Analysis

The results are indicated as the mean ± standard deviation (SD), and all measurements were repeated three times. Comparisons between two groups were performed with the Student’s *t*-test. SPSS ver. 18 (SPSS Inc., Chicago, IL, USA) was used for analysis. Statistical significance was defined as a 0.05 probability level.

## 3. Results

### 3.1. Determination of the Timing of Maximal Expression of AMP in TML Injected with L. plantarum

At the transcript level, the expression levels of the four AMPs were confirmed in samples incubated for 6, 12, 24, 36, 48, and 72 h after *L. plantarum* injection. The results confirmed that tenecin, coleoptericin, attacin, and defensin were maximally expressed 12 h (6.02-fold), 12 h (10.54-fold), 24 h (7.97-fold), and 12 h (7.915-fold), respectively (Figure 1). Unlike the other three AMPs, which were maximally expressed after 12 h, attacin showed maximum expression levels after 24 h, but the expression level was significantly higher than that of the control even after 12 h (5.77-fold). Therefore, we decided to use TML 12 h after *Lactiplantibacillus plantarum* injection for the preparation of natural preservatives.

### 3.2. Inhibition Activity of iTME against Various Microorganisms

To confirm whether iTME selectively inhibits harmful bacteria, we investigated its antibacterial activity against *E. coli*, *B. cereus*, *S. aureus*, *B. subtilis*, and *L. acidophilus* (Figure 2). Results showed that iTME had a concentration-dependent inhibitory effect on *E. coli*, *B. cereus*, and *S. aureus*, all of which are food-poisoning bacteria, and *B. subtilis*, a fermenting bacterium in Korean traditional sauces. Meanwhile, iTME had no inhibitory activity against the beneficial bacterium *L. acidophilus*.

Moreover, we investigated the antifungal activity of iTME against *A. parasiticus*, *A. flavus*, *P. anomala*, and *A. oryzae* (Figure 3). RDA confirmed that iTME had a dose-dependent antifungal activity against harmful fungi *A. parasiticus*, *A. flavus*, and *P. anomala*, as well as the beneficial fungus *A. oryzae*. The above results confirmed that iTME has an antimicrobial activity against food-poisoning bacteria and harmful aflatoxin-producing fungi that cause diarrhea, vomiting, and headaches [34]. Moreover, iTME exhibited inhibitory activity against *B. subtilis* and *A. oryzae*, which are used in Korean traditional sauce fermentation, but the inhibition level against *A. oryzae* was lower than that against harmful fungi. In addition, iTME showed antifungal activity against *P. anomala*, which is a biofilm yeast related to the reduction in quality of the sauce.

### 3.3. MIC of iTME against Food Poisoning Bacteria

The MICs of iTME against *E. coli*, *B. cereus*, and *S. aureus* detected in traditional sauces were confirmed (Table 2). As a result, OD_600_ values of 0.040 ± 0.001, 0.040 ± 0.002, 0.041 ± 0.001, and 0.040 ± 0.001 at 0, 0.8, 0.9, and 1 mg/mL of iTME, respectively, were obtained when iTME alone were added to the wells without bacteria. When 1 mg/mL of iTME was added to *E. coli*, *B. cereus*, *and S. aureus*, the OD_600_ values were 0.044 ± 0.002, 0.042 ± 0.001, and 0.041 ± 0.003, respectively, similar to the control. Accordingly, it was confirmed that the MIC was 1 mg/mL of iTME at 1 × 10^6^ CFU/mL of bacteria.

### 3.4. Stability of the Antimicrobial Activity of iTME under Various Environments

To confirm the applicability of iTME in Korean traditional sauces, varying environmental conditions of the Korean traditional sauces were inferred, and the stability in each environment was tested. We confirmed the antimicrobial activity of iTME by targeting various microorganisms that are problematic in food. And, as stability is only to determine whether or not to maintain antibacterial activity under each condition, we judged that it was sufficient to target only *E. coli*, one of the target strains already showing activity. In other papers previously reported, even if several strains were used when investigating the antimicrobial activity spectrum, only one strain was used for the stability test [36,37,38].

#### 3.4.1. High Temperature

In this study, we evaluated whether iTME was temperature-stable when used as a natural preservative in Korean traditional sauce. First, we determined the different temperature ranges to which the sauce would be exposed during distribution and sterilization, then we measured the MIC after exposing the iTME to that temperature (Table 3). When 0.8, 0.9, 1.0 mg/mL of unheated iTME was treated against 1.0 × 10^6^ CFU/mL of *E. coli*, the bacterial inhibitory concentration was 1 mg/mL, and the OD_600_ value of cultured media was 0.042 ± 0.001, which was similar to that of the control. Even when iTME was heated at 40 °C for 36 h (0.051 ± 0.002), 48 h (0.043 ± 0.001), and 72 h (0.041 ± 0.001), the inhibitory activity of 1 mg/mL of iTME was similar to that of unheated iTME. Accordingly, the antibacterial activity of iTME was stable when it was heated at 40 °C for 72 h. Moreover, the MICs of iTME heated to 100 °C for 5, 10, and 15 s against *E. coli* were 0.038 ± 0.001, 0.038 ± 0.000, and 0.042 ± 0.002, respectively. This indicates that the antimicrobial activity retained even when iTME is heated at 100 °C for 15 s. In conclusion, iTME is stable at high temperatures during the manufacture or distribution of Korean traditional sauce.

#### 3.4.2. pH

We evaluated whether iTME was stable at varying pH (Table 3). Moreover, the MIC was measured after exposing the iTME to that pH. When iTME (1 mg/mL) of pH 7 was treated with 1.0 × 10^6^ CFU/mL of *E. coli*, the bacterial inhibitory concentration was 1 mg/mL, and the OD_600_ value of bacteria cultured media was 0.042 ± 0.001. Meanwhile, the OD_600_ values of the samples treated with 1 mg/mL of iTME at 4, 5, and 6 were 0.040 ± 0.001, 0.041 ± 0.000, and 0.036 ± 0.002, respectively. These values are similar to those at pH 7. These results suggest that the antimicrobial activity of iTME can be maintained under various pH conditions.

#### 3.4.3. Salinity

We assessed whether iTME is stable to salinity. Various salinity ranges to which the sauce could be exposed were determined (Table 1), and the MIC was measured after exposing the iTME to these salinities. Results showed that the salinity of Ganjang and Gochujang were 19.33 ± 1.68% and 6.96 ± 0.30%, respectively (Table 3). Subsequently, salinity stability was measured at 5%, 7.5%, 10%, 15%, 20%, and 25%. When iTME was treated with 1.0 × 10^6^ CFU/mL of *E. coli* in normal medium, the bacterial inhibitory concentration was 1 mg/mL, and the OD_600_ value was 0.042 ± 0.001. Moreover, the OD_600_ value of bacteria cultured media when treated with 1 mg/mL of iTME under 5%, 7.5%, 10%, 15%, 20%, and 25% salinity was 0.041 ± 0.001, 0.045 ± 0.002, 0.042 ± 0.001, 0.042 ± 0.000, 0.041 ± 0.000, and 0.042 ± 0.001, respectively. These values were similar to that in normal media. These results suggest that the antimicrobial activity of iTME can be maintained even under varying salinities.

### 3.5. Inactivation of the Antibacterial Activity of iTME under Human Proteolytic Enzymes

We also confirmed whether the antibacterial activity of iTME under human intestinal protease was maintained (Table 4). When untreated iTME was added to 1.0 × 10^6^ CFU/mL of *E. coli*, *B. cereus*, and *S. aureus*, the bacterial inhibitory concentration was 1 mg/mL, and the OD_600_ value of the bacterial culture media was 0.042 ± 0.001. Meanwhile, iTME (1 mg/mL) reacted with proteolytic enzymes for 5, 10, 20, 30, and 60 min did not exhibit inhibitory activities against *E. coli*, *B. cereus*, and *S. aureus* (Table 4). These results indicate that iTME would lose its inhibitory activity when introduced into the intestine.

### 3.6. Analysis of the Antimicrobial Activity of iTME in Korean Traditional Soy Sauce Inoculated with Harmful Bacteria

The results mentioned above confirmed that iTME exhibited the same antimicrobial activity when applied to Ganjang, in which harmful bacteria were present. At first, Ganjang inoculated with 1 × 10^5^ CFU/mL of *E. coli*, *B. cereus*, *S. aureus* was cultured at 4, 18, and 37 °C for 10 d. Bacterial growth was then observed (Table 5). In the case of preservative-free and *E. coli*-free Ganjang, *E. coli* was not detected even after 10 d at 37 °C, which is the optimum growth temperature for *E. coli*. However, when *E. coli* (1.87 × 10^3^ CFU/mL) was inoculated, the number of *E. coli* in the Ganjang increased by approximately 1.5 fold (2.86 × 10^3^ CFU/mL) after 10 d. When *E. coli* and iTME were simultaneously added in Ganjang and stored at 4, 18, and 37 °C, bacterial colonies were not detected from 2 to 10 d. Similarly, *B. cereus* and *S. aureus* colonies were not detected on the 2 d after of all temperature treatment groups. Moreover, no colonies were observed 2 d post-inoculation of iTME, even when three types of food poisoning bacteria were inoculated at a concentration of 10^3^ or higher. Therefore, iTME can extend the shelf life of Ganjang as it inhibits bacterial growth and removes harmful microorganisms.

## 4. Discussion

Problems such as food poisoning bacteria, aflatoxin, and biofilm formation are emerging in the sauces we eat every day in our daily life. To solve this problem, chemical preservatives are being used, but there is a feeling of objection from consumers [6]. Insects recognize microbial wall components during microbial infection, express various types of AMPs, and secrete them out of the cell [39]. AMPs have target-specific natural antimicrobial activities and the possibility of inducing resistant strains is low since AMPs destroy the cell wall of bacteria [10]. In previous reports, insect-derived AMPs have been industrially applied by expressing or synthesizing every single peptide, but only one peptide has a narrow spectrum of antimicrobial activity [39]. Therefore, in this study, the production of various AMPs in the insect body was induced through the injection of edible lactic acid bacteria, and the extract (iTME) was prepared with an edible solvent. The maximum expression time of AMPs was confirmed after lactic acid bacteria injection (Figure 1). In general, the expression levels of AMPs are very low under normal conditions but are significantly increased when the immune system is activated [40]. In this study, it was confirmed that AMP was overexpressed in TML at 12 to 24 h after bacterial injection compared to normal larvae. These results were consistent with those previously reported that the time of maximal expression of tenecin, attacin, defensin, coleoptericin, and cecropin in TML injected with *E. coli* and *S. aureus* was 24, 24, 12, 24, and 24 h, respectively [35]. iTME showed antibacterial activity against food poisoning bacteria. In the case of antifungal activity, it was shown in the aflatoxin-producing strain. In addition, it showed antifungal activity not only in biofilm yeast but also in fermented strains of Korean traditional sauces. Fermented foods become over-fermented as the fermentation process continues, and the taste and aroma of over-fermented foods change, while biofilm yeast grows, which induces spoilage [41,42]. The inhibitory activity of iTME against fermenting strains and biofilm yeast could prevent the over-fermentation and spoilage that may occur during the distribution process. Therefore, when using iTME in the production of Meju or Korean traditional sauce, it is recommended to apply it after fermentation and before packaging. Although additional experiments using diverse beneficial and harmful strains are needed, the results of this study suggest that iTME can be used to inhibit harmful bacteria and fungi and prevent over-fermentation by controlling the time of incubation and amount of iTME.

The MIC of iTME was measured to be 1 mg/mL when the number of bacteria was 1.0 × 10^6^ CFU/mL. In addition, in an experiment on the stability of iTME, it was found that iTME did not lose its antibacterial activity at high temperature and at various pH and salinities. MICs of sorbic acid, a representative synthetic preservative used in food, against *E. coli*, *B. cereus*, and *S. aureus* were 4, 2, and 2 mg/mL, respectively, and those of nisin were 0.5, 2, and 32 mg/mL, respectively [42,43]. The MIC of grapefruit seed extract (GSE) against *E. coli* was 4 mg/mL [44]. These studies show that iTME had a lower MIC than sorbic acid, GSE, and nisin. As for the stability as food preservatives, sorbic acid has been reported to have a very stable antimicrobial activity at pH 2–7, after heating at 85 °C for 2 h, or at a high salt condition [45]. Nisin, a natural preservative derived from bacteria and sold as a commercial product, has also been shown to have antimicrobial activity at pH 2–6 and even when heated to 121 °C [46]. In addition, GSE also retains its antibacterial activity at pH 4–10 and 150 °C [44]. These results suggest that the shelf life may be extended if the antimicrobial activity is maintained under various processing and storage conditions. Combining the above results, iTME had stronger antimicrobial activity than other food preservatives and showed a more stable activity against food-poisoning bacteria. Therefore, we suggest that iTME may effectively extend the shelf life of various foods containing Korean traditional sauces. A previous study reported that bacteriocin, a representative AMP used in food, is hydrolyzed when reacted with the human proteolytic enzyme and loses its antimicrobial activity. Therefore, the antimicrobial activity of bacteriocin does not remain and is used as a food preservative [11,46]. iTME showed a result of losing antibacterial activity by human-derived trypsin and chymotrypsinin this study. Therefore, when ingested, iTME may not remain in the intestine due to proteolytic enzymes, nor does it exhibit antibacterial activity against *L. acidophilus*, a human intestinal bacterium (Figure 2). Therefore, iTME is predicted to be safe for the human body. In addition, its safety has been proven through single-dose toxicity, subchronic toxicity, and genotoxicity tests based on good laboratory practice guidelines when registered as a new food raw material [47,48]. In the results of the experiment in which iTME was added to the Korean traditional soy sauce inoculated with each food poisoning bacteria, no bacteria were detected after 2 d had elapsed. This indicates that iTME effectively inhibits food poisoning bacteria without losing its antibacterial activity even at the distribution or storage temperature of Korean traditional soy sauce with high salinity and low pH. Thus, iTME is not only a potential novel natural food preservative but also shows the possibility of using it as a preservative for various items used in daily life.

## 5. Conclusions

In this study, the possibility of using iTME which contains a large number of various types of AMPs as a preservative in food storage and distribution was confirmed. It had inhibitory activity against food poisoning bacteria, acid film-forming yeast, and aflatoxin-producing fungi, which are problematic in Korean traditional sauce. In addition, it was predicted that the activity would be very stable under various environments in the storage and distribution process and safe for the human body. Therefore, iTME is expected to be used as a natural preservative to prevent food poisoning by microorganisms in various kinds of food, including Korean traditional sauces.

## Figures and Tables

**Figure 1 insects-13-00381-f001:**
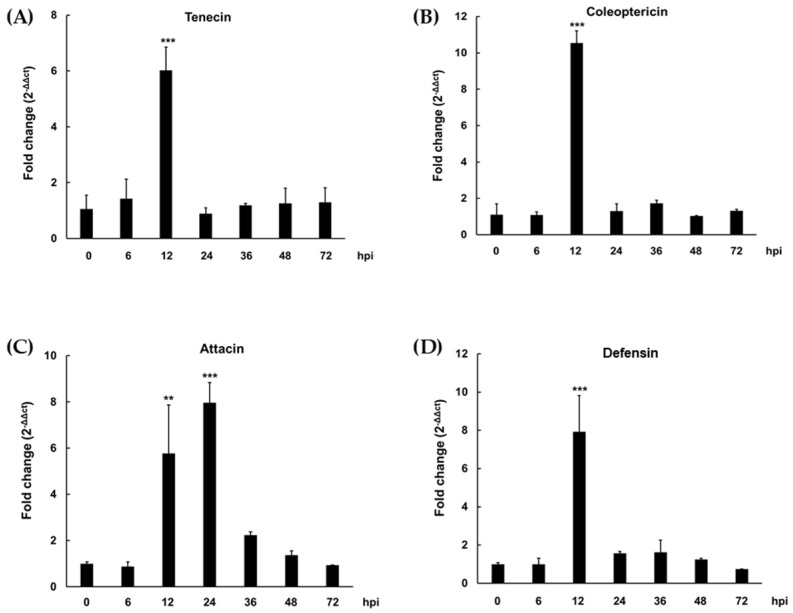
Gene expression of tenecin (**A**), coleoptericin (**B**), attacin (**C**), and defensin (**D**) in *Tenebrio molitor* larvae (TML). Values are reported as mean ± standard deviation of triplicate experiments. ** *p* < 0.01 and *** *p <* 0.001 indicate significant difference between the 0 h and 6, 12, 24, 36, 48, and 72 h post infection (hpi), respectively.

**Figure 2 insects-13-00381-f002:**
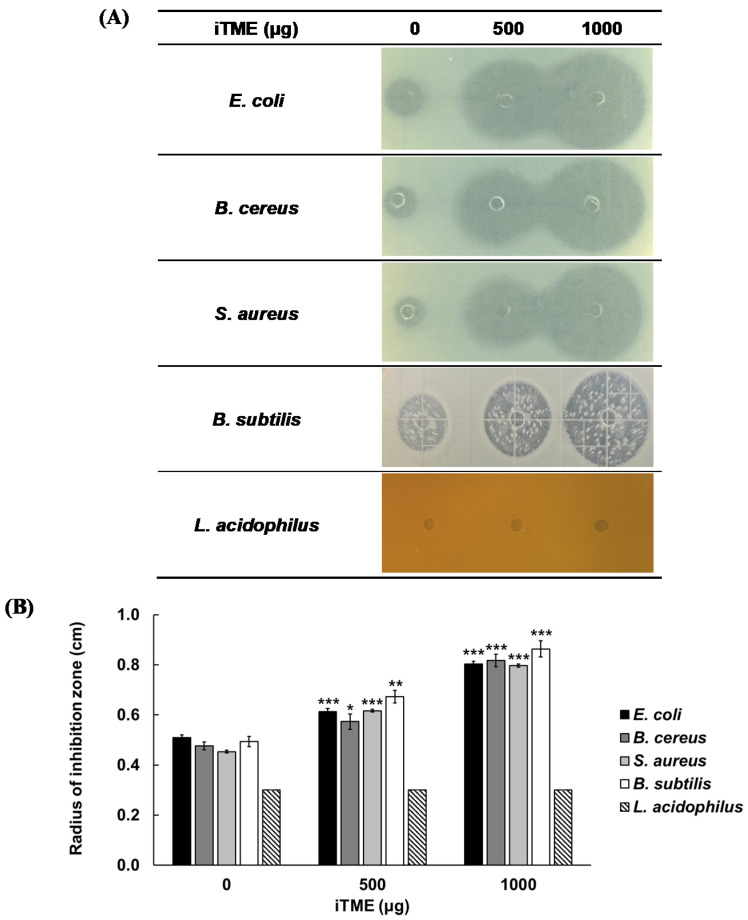
Evaluation of the antibacterial activities of immunized *Tenebrio molitor* larvae extract (iTME) against harmful bacteria (*Escherichia coli*, *Bacillus cereus*, and *Staphylococcus aureus*), and beneficial bacteria (*Bacillus subtilis*, *Lactobacillus acidophilus*) using radial diffusion assay (RDA) (**A**) and by measuring the radius of inhibition zone (**B**), Values are reported as mean ± standard deviation of triplicate experiments. * *p <* 0.05, ** *p <* 0.01, and *** *p <* 0.001 represent significant differences between 0 µg and 500, 1000 µg, respectively.

**Figure 3 insects-13-00381-f003:**
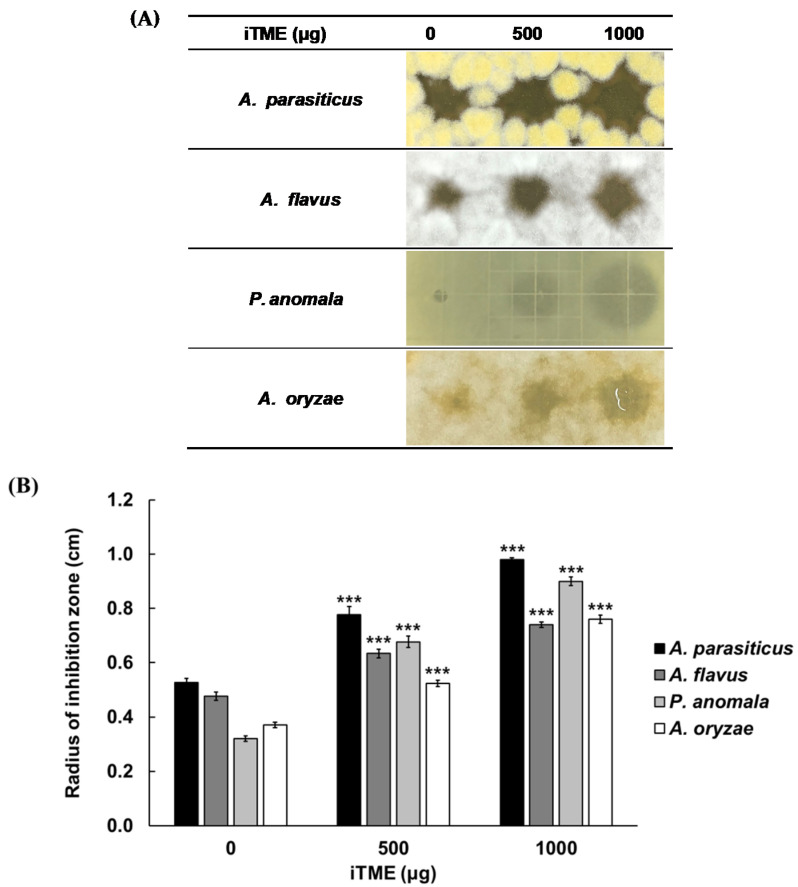
Evaluation of the antifungal activities of iTME against harmful fungi (*Aspergillus parasiticus*, *A. flavus*, *Pichia anomala*), and beneficial fungi (*A. oryzae*) using RDA (**A**) and by measuring the radius of inhibition zone (**B**). Values are reported as mean ± standard deviation of triplicate experiments. *** *p <* 0.001 represent significant differences between 0 µg and 500, 1000 µg, respectively.

**Table 1 insects-13-00381-t001:** The pH and salinity of commercially-available Ganjang and Gochujang.

	Ganjang	Gochujang
pH	4.74 ± 0.09	5.04 ± 0.07
Salinity (%)	19.33 ± 1.68	6.96 ± 0.30

**Table 2 insects-13-00381-t002:** Minimal inhibitory concentration (MIC) of immunized *Tenebrio molitor* larva extract against *E. coli*, *B*. *cereus*, and *S. aureus*.

iTME (mg/mL)	0	0.8	0.9	1
*-*	0.040 ± 0.001	0.040 ± 0.002	0.041 ± 0.001	0.040 ± 0.001
*E. coli*	1.159 ± 0.121	1.152 ± 0.045	0.208 ± 0.014	0.044 ± 0.002
*B. cereus*	0.862 ± 0.092	0.646 ± 0.076	0.267 ± 0.015	0.042 ± 0.001
*S. aureus*	1.095 ± 0.071	1.103 ± 0.047	0.164 ± 0.005	0.041 ± 0.003

0, 0.8, 0.9, and 1 mg/mL of iTME were mixed with 1 × 10^6^ CFU/mL of *E. coli*, *B. cereus*, *and S. aureus* and incubated at 30 °C or 37 °C for 24 h. Subsequently, the OD_600_ value was measured. iTME: immunized *Tenebrio molitor* larva extract.

**Table 3 insects-13-00381-t003:** Stability analysis of iTME under varying temperature, pH, and salinity.

*E. coli* iTME (mg/mL)			− 0	+ 0.8	+ 0.9	+ 1
Heating time	40 °C (h)	0	0.041 ± 0.001	1.131 ± 0.142	0.257 ± 0.012	0.042 ± 0.001
		36	0.040 ± 0.000	1.148 ± 0.087	0.195 ± 0.014	0.051 ± 0.002
		48	0.042 ± 0.001	1.195 ± 0.098	0.210 ± 0.010	0.043 ± 0.001
		72	0.041 ± 0.001	1.145 ± 0.081	0.187 ± 0.009	0.041 ± 0.001
	100 °C (s)	5	0.040 ± 0.000	1.200 ± 0.104	0.174 ± 0.014	0.038 ± 0.001
		10	0.040 ± 0.000	1.317 ± 0.124	0.147 ± 0.024	0.038 ± 0.000
		15	0.041 ± 0.001	1.166 ± 0.099	0.121 ± 0.041	0.042 ± 0.002
		4	0.041 ± 0.001	1.138 ± 0.045	0.217 ± 0.047	0.040 ± 0.001
pH		5	0.041 ± 0.002	1.097 ± 0.084	0.225 ± 0.035	0.041 ± 0.000
		6	0.042 ± 0.001	1.126 ± 0.091	0.237 ± 0.087	0.036 ± 0.002
		7	0.042 ± 0.000	0.042 ± 0.001	0.041 ± 0.001	0.042 ± 0.000
		0	0.042 ± 0.000	0.041 ± 0.001	0.041 ± 0.001	0.040 ± 0.001
		5	0.042 ± 0.001	0.635 ± 0.092	0.585 ± 0.024	0.041 ± 0.001
		7.5	0.041 ± 0.000	0.873 ± 0.052	0.581 ± 0.068	0.045 ± 0.002
Salinity (%)		10	0.041 ± 0.001	0.646 ± 0.089	0.267 ± 0.042	0.042 ± 0.001
		15	0.043 ± 0.002	0.710 ± 0.052	0.244 ± 0.023	0.042 ± 0.000
		20	0.042 ± 0.002	0.823 ± 0.038	0.243 ± 0.012	0.041 ± 0.000
		25	0.042 ± 0.001	0.748 ± 0.047	0.242 ± 0.027	0.042 ± 0.001

After pretreating iTME under the above conditions or changing the conditions to 0, 0.8, 0.9, and 1 mg/mL of iTME mixed with 1 × 10^6^ CFU/mL of *E. coli* and incubated at 37 °C for 18 h, the OD_600_ value was measured. iTME: immunized *Tenebrio molitor* larva extract.

**Table 4 insects-13-00381-t004:** Analysis of the antimicrobial activity of iTME under proteolytic enzymes.

Bacteria Reaction Time (Min)	-	*E. coli*	*B. cereus*	*S. aureus*
0	0.040 ± 0.001	1.052 ± 0.152	1.014 ± 0.087	1.151 ± 0.124
5	0.041 ± 0.002	1.125 ± 0.102	0.812 ± 0.084	1.043 ± 0.098
10	0.040 ± 0.000	1.248 ± 0.108	0.931 ± 0.096	1.012 ± 0.125
20	0.042 ± 0.001	1.302 ± 0.138	0.948 ± 0.090	0.957 ± 0.082
30	0.042 ± 0.000	1.147 ± 0.085	1.027 ± 0.153	0.995 ± 0.097
60	0.042 ± 0.001	1.217 ± 0.257	0.985 ± 0.185	1.092 ± 0.127

iTME (1 mg/mL) was mixed with 1 × 10^6^ CFU/mL of *E. coli*, *B. cereus*, *and S. aureus* and incubated at 30 °C or 37 °C for 24 h. Subsequently, the OD_600_ value was measured. iTME: immunized *Tenebrio molitor* larvae extract.

**Table 5 insects-13-00381-t005:** Evaluation iTME as a preservative of Korean traditional soy sauce.

iTME (1 mg/mL)	Temperature (°C)	Bacteria	Storage Time (d)					
			0	2	4	6	8	10
−	37	*E. coli* (CFU/mL)	0	0	0	0	0	0
−	37		1870 ± 72	1680 ± 65	2150 ± 74	2480 ± 89	2520 ± 41	2860 ± 87
+	4		1562 ± 54	0	0	0	0	0
+	18		1847 ± 49	0	0	0	0	0
+	37		2012 ± 86	0	0	0	0	0
−	37	*B. cereus* (CFU/mL)	0	0	0	0	0	0
−	37		4852 ± 105	3420 ± 47	5064 ± 190	4890 ± 187	5385 ± 215	5597 ± 301
+	4		4181 ± 214	0	0	0	0	0
+	18		3988 ± 171	0	0	0	0	0
+	37		4773 ± 114	0	0	0	0	0
−	37	*S. aureus* (CFU/mL)	0	0	0	0	0	0
−	37		160 ± 7	350 ± 12	470 ± 9	620 ± 24	2830 ± 57	3541 ± 107
+	4		80 ± 4	0	0	0	0	0
+	18		110 ± 14	0	0	0	0	0
+	37		230 ± 19	0	0	0	0	0

After simultaneous treatment of 1 × 10^5^ CFU/mL of *E. coli* (A), *B. cereus* (B), or *S. aureus* (C), with 1 mg/mL of iTME in Ganjang without preservative, the colony forming unit (CFU) was measured after incubation at 4, 18, or 37 °C for 10 d. The number of colonies was counted after culturing 100 μL of the sample in selective medium (*E. coli*, MacConkey agar media; *B. cereus*, mannitol egg yolk polymyxin agar; *S. aureus*, mannitol salt agar with egg yolk) every 2 d. All treatments were stored at the optimal growth temperature of each bacterial strain (*E. coli*, 37 °C; *B. cereus*, 30 °C; *S. aureus*, 37 °C). iTME: immunized *Tenebrio molitor* larvae extract.

## Data Availability

The data presented in this study are available on reasonable request from the corresponding author.

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
