# Peer review of "Potential of Antimicrobial Peptide-Overexpressed Tenebrio molitor Larvae Extract as a Natural Preservative for Korean Traditional Sauces"

_insects, 2022, doi:10.3390/insects13040381_

Round 1

Reviewer 1 Report

The authors performed a very interesting work, rich with information. This work should be considered of high value, although I detected many issues that should be addressed and clarified.

The authors should write results, discussion and conclusion separately.

Please change the following keywords, as they are already in the title: Korean traditional sauces; natural preservative; Tenebrio molitor

ABSTRACT

Line 20, write “Aspergillus parasiticus”

Line 22 write “Furthermore”

INTRODUCTION

Line 26 write “hung” not “hanged”

Line 54 write “traditional sauces”

Line 61, please add a reference to this sentence “Currently, AMPs are used in various fields such as antibiotics, anticancer drugs, cosmetic materials, and food.” (for example doi: 10.3389/fcimb.2021.668632)

Line 69 “Insects produce a variety of AMPs for defense against external pathogens, and they 69 harbor more than 150 different AMP genes” please add more recent references (for example https://doi.org/10.1038/s41598-020-74017-9, https://doi.org/10.1007/s00018-021-03784-z, https://doi.org/10.3390/cimb44010001)

MATERIAL AND METHODS

Line 105: “induce the production of AMP” the word AMP should be written in the plural

Line 106: “left” it is advisable to replace the word with a more appropriate synonym

Line 108: it is not specified which acetic acid has been used (please add the brand)

Line 160 write “the pH of these products was adjusted”

Line 199 please specify the origin of the sequence of T. molitor transcripts (attacin, defensin, tenecin, and coleoptericin)

Line 121 According to the guideline reported in “Minimum Information Required for Publication of Quantitative Real-Time PCR experiments (MIQE) (Bustin et al., 2009) multiple reference genes are needed. Why did the authors use just actin? Please repeat the experiment with at least one more gene, alternatively, explain the previous validation of this gene. Moreover, the authors explain that the analysis was performed through the delta delta CT method, but, in order to use these equations, the efficiencies of the amplicons must be approximately equal and between the values of 0.8 and 1. Please explain.

Please write the origin of the sequence used for the qPCR, for example the accession number. Indeed, the sequences of actin and defensin primers seem to be incorrect

Although the authors wrote the paragraph “2.7. Analysis of the antibacterial activity of iTME under human proteolytic enzymes” in the result section there are not related experiments.

It is interesting the evident stability of iTME antimicrobial activity under various environments.  Why has this stability of antimicrobial activity under pressure, pH and salinity been tested only against E. coli?

Please add a paragraph in which the statistical analysis performed is reported.

RESULTS

Line 268: a value of 0.041± 0.001 is given in the text for the concentration 0 mg/mL while the value of 0.040 ± 0.001 is shown in the table.

The images in Figure 3 (A) for the inhibition haloes of A. parasiticus and A. flavus relative to the difference between halo 0 and the other two do not agree quantitatively and proportionally with the significant difference that is plotted in Figure 3 (B).

FIGURE CAPTION

In figure 1 write “defensin”.

In figure 2 write “Bacillus” not “Bacillis”

In figure 2 write “Pichia” not “Pischia”

Please report the statistical analysis the authors performed.

Author Response

 We submit the response to the reviewer’s comments as an attached file.

Reviewer 2 Report

I have only a few suggestions:

It is not appropriate to use non-unique abbreviations in the title. AMP has many meanings.

76-84 I suggest moving this to Methods:  „Therefore, in this study, we utilized insect-derived AMPs as  a sauce preservatives. An extract containing AMPs, which exhibits a wide range of anti- bacterial activity and is inexpensive, was prepared using Tenebrio molitor larvae (meal- worm, TML), an edible insect. Since the preparation of Meju with TML has been previously reported [29], we assumed that this extract would not deteriorate the taste of Gan- 80 jang. To prepare an extract containing a large amount of TML-derived AMPs that can be used as a preservative in Korean traditional sauces, we injected lactic acid bacteria into TML and prepared the extract with various overexpressed AMPs using glacial acetic acid. Finally, to confirm the potential of immunized Tenebrio molitor larvae extract (iTME) as a sauce preservative, we analyzed the antimicrobial activity and stability of iTME.

195-202 I suggest moving this to Introduction: „To prevent contamination caused by toxin-producing microorganisms in Korean traditional sauces, here we used TML, an edible insect in Korea, to manufacture a natural preservative. Generally, AMP has a wide range of natural antimicrobial activities; since AMP destroys the cell wall of bacteria, the possibility of inducing resistant strains is low  [34]. Therefore, to prepare a natural preservative that can be applied to food, Lactiplanti bacillus plantarum, an edible gram-positive lactic acid bacterium, was selected as an immune inducer for inducing AMP expression in the TML and injected into the abdomen of  the larvae. The use of TML for the development of more effective preservatives is advan- tageous, as large number of various AMPs are expressed.

304 – 314 In the text, chapters: 3.4.2. a 3.4.3. there are references to Table 1, please check if the references shouldn’t be to Table 3, which presents the results.

In the Results, the topic is very little discussed.  According to the “Instructions for Authors“: Authors should discuss the results and how they can be interpreted in perspective of previous studies and of the working hypotheses. The findings and their implications should be discussed in the broadest context possible, and limitations of the work highlighted...

Although it is obvious that this is very innovative and primary research, I suggest extending the discussion as possible in the Results and changing the chapter title to “3. Results and Discussion“.

Author Response

(The authors gave the same response as above.)

Round 2

Reviewer 1 Report

The authors responded satisfactorily to all the questions.

I suggest to include in the manuscript the explanation they reported concerning the experiment exclusively on E. coli "We confirmed the antimicrobial activity of iTME by targeting various microorganisms that are problematic in food. And, as stability is only to determine whether or not to maintain antibacterial activity under each condition, we judged that 
it was sufficient to target only E. coli, one of the target strains already showing activity. In other papers previously reported, even if several strains were used when investigating the antimicrobial activity spectrum, only one strain was used for the stability test (Abebe et al., 2020; Tan et al., 2015; Phupiewkham et al., 2010)."

Author Response

Thank you for your time in handling our manuscript. We have improved our manuscript according to the reviewers' suggestions. In the manuscript file, we have used the track function to mark the changes through the 2nd revision using the manuscripts revised by the 1st revision. The response to the specific comments is as follows:

<Revision List>

The authors responded satisfactorily to all the questions.

  1. I suggest to include in the manuscript the explanation they reported concerning the experiment exclusively on E. coli "We confirmed the antimicrobial activity of iTME by targeting various microorganisms that are problematic in food. And, as stability is only to determine whether or not to maintain antibacterial activity under each condition, we judged that it was sufficient to target only E. coli, one of the target strains already showing activity. In other papers previously reported, even if several strains were used when investigating the antimicrobial activity spectrum, only one strain was used for the stability test (Abebe et al., 2020; Tan et al., 2015; Phupiewkham et al., 2010)."

⟶ Answer: According to the review’s comments, we added "We confirmed the antimicrobial activity of iTME by targeting various microorganisms that are problematic in food. And, as stability is only to determine whether or not to maintain antibacterial activity under each condition, we judged that it was sufficient to target only E. coli, one of the target strains already showing activity. In other papers previously reported, even if several strains were used when investigating the antimicrobial activity spectrum, only one strain was used for the stability test (Abebe et al., 2020; Tan et al., 2015; Phupiewkham et al., 2010)." to the section of “3.4. Stability of the antimicrobial activity of iTME under various environments (Line 306-312)”.